# LC3-associated phagocytosis in macrophage responses to *Paracoccidioides* spp.

**Getúlio Pereira de Oliveira Júnior**[1,2], **Herdson Renney de Sousa**[3], **Kaio César de Melo Gorgonha**[3], **Lara Laís Montalvão Tomaz**[3], **Tatiana Karla dos Santos Borges**[3], **Kellyanne Teixeira Rangel**[2], **Scott Fabricant**[3], **Fernanda Koser Gustafson**[3], **Lucas Friaça Albuquerque**[3], **Angelo Rossi Neto**[4], **Fabián Andrés Hurtado**[5], **Hugo Costa Paes**[3], **Arturo Casadevall**[6], **Ildinete Silva-Pereira**[5], **Maria Sueli Soares Felipe**[2,5], **Patrícia Albuquerque**[7,8], **André Moraes Nicola**[3,9,10]/+

[1]Northeastern University, Barnett Institute for Chemical & Biological Analysis, Boston, MA, USA
[2]Universidade Católica de Brasília, Programa de Pós-Graduação em Ciências Genômicas e Biotecnologia, Brasília, DF, Brasil
[3]Universidade de Brasília, Faculdade de Medicina, Brasília, DF, Brasil
[4]Universidade Católica de Brasília, Escola de Medicina, Brasília, DF, Brasil
[5]Universidade de Brasília, Instituto de Ciências Biológicas, Brasília, DF, Brasil
[6]Johns Hopkins University, Johns Hopkins Bloomberg School of Public Health, Baltimore, MD, USA
[7]Universidade de Brasília, Faculdade de Ciências e Tecnologias da Saúde, Brasília, DF, Brasil
[8]Cornell University, Weill Cornell Medicine, Jill Roberts Institute for Research in Inflammatory Bowel Disease, New York, NY, USA
[9]Rockefeller University, Laboratory of Molecular Immunology, New York, NY, USA
[10]Universidade de São Paulo, Instituto Nacional de Ciência e Tecnologia Fungos Patogênicos Humanos, São Paulo, SP, Brasil

**BACKGROUND** Paracoccidioidomycosis (PCM) is a systemic infection that is endemic to Latin America, caused by thermodimorphic fungi from the *Paracoccidioides* genus. These fungi are facultative intracellular parasites of macrophages. LC3-associated phagocytosis (LAP), a non-canonical form of autophagy, plays a critical role in the response of these phagocytes to similar pathogens.

**OBJECTIVES** In this study, we investigated the role of LAP in the macrophage responses to *Paracoccidioides* spp.

**METHODS** We detected LAP in macrophages infected with *Paracoccidioides* spp by immunofluorescence microscopy with antibodies to LC3. Piceatannol and diphenyleneiodonium chloride (DPI), respectively Syk and nicotinamide adenine dinucleotide phosphate oxidase (NADPH) inhibitors, were used to understand the role their pathways played. To determine the function of LAP, we targeted *ATG5*, a key autophagy gene, by RNA interference.

**FINDINGS** We observed LC3 recruitment to phagosomes containing *Paracoccidioides* spp. in RAW264.7 and J774.16 cell lines and in bone marrow-derived macrophages. *ATG5* RNA interference reduced the antifungal activity of J774.16 cells, highlighting the importance of LC3 recruitment for effective fungal control. Interestingly, pharmacological inhibition of Syk kinase and NADPH oxidase pathways, essential for LAP against *Aspergillus fumigatus* and *Candida albicans*, did not impair LAP against *P. brasiliensis*.

**MAIN CONCLUSIONS** This suggests distinct triggering mechanisms, possibly due to differences in the fungal cell surface composition. These findings suggest that LAP plays a significant role in the host defense against *Paracoccidioides* spp. and may represent a promising target for host-directed PCM therapies.

Key words: autophagy - LC3-associated phagocytosis (LAP) - macrophage - *Paracoccidioides* - fluorescence microscopy

The *Paracoccidioides* genus includes thermodimorphic fungi that can be isolated from the soil,[1] especially around armadillo burrows.[2] Upon inhalation by humans, they can cause paracoccidioidomycosis (PCM) — one of the most prevalent systemic mycoses in Latin America. [3] PCM manifests as either chronic disease localized to mucosae and lungs or acute systemic infection, with high fungal burden in lymph nodes and the spleen.[4] The disease poses significant public health burden, with annual incidence as high as 9.4 per 100,000 and a case-fatality rate of approximately 6% in endemic regions.[5]

Macrophages play a central role in the immune response to *Paracoccidioides* spp., with their functional states (pro-inflammatory M1 versus anti-inflammatory M2) influencing disease outcomes.[6] Activated M1 macrophages secrete cytokines such as IL-12 and produce

Financial support: AMN was funded by FAP-DF (awards 0193.001048/2015-0193.001561/2017 and the CNPq grants 437484/2018-1 and 405934/2022-0); MSSF was supported by FAP-DF/PRONEX (award 193.001.533/2016); GPOJ was supported by a scholarship from CAPES (number 150510/2017-9). GPOJ and HRS contributed equally to this work.

+ Corresponding author: amnicola@unb.br | ⓘ https://orcid.org/0000-0001-8656-5835

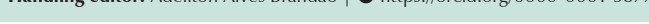

**Handling editor:** Adeilton Alves Brandão | ⓘ https://orcid.org/0000-0001-5877-607X

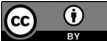

nitric oxide, which are critical for fungal clearance.[7] However, *Paracoccidioides* infections are sometimes associated with immune dysregulation, leading to granulomatous inflammation and impaired fungal control that result in active disease.[8]

Current PCM treatments, including sulfonamides, amphotericin B, and azole derivatives, are lengthy, costly, and often associated with adverse effects or relapse.[9] This underscores the need for novel therapeutic approaches, including strategies targeting host immune mechanisms.[10]

Autophagy, a conserved eukaryote homeostatic process,[11,12,13] plays a key role in host immunity against intracellular pathogens including fungi.[14,15] Among its variants, LC3 associated phagocytosis (LAP) is triggered during macrophage responses to pathogens such as *Aspergillus fumigatus*,[16,17] *Cryptococcus neoformans*,[18,19] *Candida albicans*,[18,20,21] *Histoplasma capsulatum*,[22,23] *Talaromyces marneffei*,[24,25] and *Saccharomyces cerevisiae*-derived zymosan.[26] Given its importance in other fungal infections, understanding LAP's role in the immune response to *Paracoccidioides* spp. could reveal novel therapeutic targets.

In this study, we investigated whether macrophages use LAP to respond to *Paracoccidioides* spp. and explored the signaling pathways involved. Our findings provide insights into the mechanisms underpinning macrophage antifungal responses and highlight LAP as a potential target for host-directed therapies.

## MATERIALS AND METHODS

*Fungal and cell cultures* - The Pb18 and Pb01 isolates of *P. brasiliensis* and *P. lutzii*, respectively, were maintained in Fava-Netto's medium (1% w/v peptone, 0.5% w/v yeast extract 0.3% w/v proteose peptone, 0.5% w/v beef extract, 0.5% w/v NaCl, 4% w/v glucose, and 1.4% w/v agar, pH 7.2), at 37ºC. Cultures no older than five days from the last passage were used for experiments. For fungicidal activity experiments, fungal colony forming units (CFUs) were counted by plating in brain-heart infusion (BHI) agar supplemented with horse serum and *P. brasiliensis* strain 192 conditioned medium. All these culture conditions are conducive to the yeast phenotype, which we used for all experiments.

*Cell lines* - The mouse macrophage cell lines RAW 264.7, and J774.16 were used for the detection of LAP *in vitro*. HEK 293T, and J774.16 were used for transfection and transduction assays, respectively. Cells were kept in 100-mm Petri dishes with supplemented Dulbecco's Modified Eagle's Medium (DMEM) with 1% non-essential amino-acid solution and 10% of fetal bovine serum (FBS) and incubated at 37ºC and 5% $CO_2$.

*Animals and primary cells* - C57BL/6 mice were bred at the Animal Center of the University of Brasília Institute of Biological Sciences with food and water *ad libitum*. All procedures were performed in accordance with national and institutional guidelines for animal care and were approved by the university's Institutional Animal Care Use Committee (Proc. UnB Doc 52657/2011). Bone marrow-derived macrophages were generated from bone marrow cells from six- to 12-week old C57BL/6 mice as

previously described,[27] a condition that results in cultures with approximately 80% macrophages.[28] Briefly, 2 x $10^6$ bone marrow cells were plated on non-tissue treated 100-mm Petri dishes with RPMI 1640 supplemented with 10% heat-inactivated FBS (Thermo Fisher), 50 µg/mL gentamicin, 50 µM 2-mercaptoethanol (Sigma-Aldrich) and 20 ng/mL recombinant GM-CSF (Peprotech). The cultures were incubated for eight days at 37ºC in a humidified 5% $CO_2$ atmosphere. On day 3, 10 mL of fresh complete RPMI was added to the culture. Half of the medium was removed on day 6 and new complete RPMI was added. Attached bone marrow derived macrophages (BMMs) were collected on day 8 with TrypLE™ Express (Thermo Fisher). Cell viability was consistently > 95%, as measured by trypan blue exclusion.

*Production of ATG5 shRNA lentiviral vectors* - For the transfection assay we used the third generation lentiviral packaging vectors pRSV-Rev, pMDLg/pRRE and pMD2.G. The pLK0.1 lentiviral vector was used to transduce shRNAs targeting murine *ATG5*. As negative control, we used a similar vector encoding an shRNA that targets the enhanced green fluorescent protein (EGFP) gene, a sequence that is not present in the cell. Plasmid expansion was performed in competent *Escherichia coli* (Omnimax T1, Thermo Fisher) in lysogeny broth (LB) with 100 µg/mL ampicillin. Plasmids were purified with the GenElute Plasmid DNA Miniprep Kit (Sigma-Aldrich), quantified using the Qubit fluorometer (Thermo Fisher) and stored at -20ºC.

HEK 293T cells were trypsinized and harvested at 90-95% confluence. Cells were reseeded at a concentration of 3.75 x $10^5$ per mL onto a six-well plate containing 2 mL of supplemented DMEM per well. A mix containing OptiMEM media, Lipofectamine 2000 (Invitrogen), and the assembly media containing the packing plasmids plus the pLKO.1 vectors encoding each shRNA were added to each well. After 6 h of incubation, 2 mL of supplemented DMEM were added to the cells, and the supernatant was collected after 12 and 24 h. Supernatants were centrifuged at 200 x g for 5 min to remove dead cells and debris, and the resulting supernatant was centrifuged again at 20.000 x g for 90 min at 4ºC. Pellets containing lentivirus were resuspended and stored at -80ºC.

*J774.16 cell transduction with ATG5 shRNA lentivirus* - After J774.16 cells reached 95% confluence, they were harvested by trypsinization and counted. They were reseeded onto a 96-well plate containing supplemented DMEM, at $10^4$ cells/well. In the following day, the cell culture medium was exchanged for supplemented DMEM with 8 µg/mL hexadimethrine bromide (Polybrene), and 5, 50 or 100 µL of lentivirus harboring each shRNA were added to the cells. In the next day, the medium was exchanged for fresh supplemented DMEM, and after one more day, transduced cells were selected with puromycin at 0.5 µg/mL (Thermo Fisher). After 48 h of selection, untransduced dead cells were removed from the supernatant. For the second round of selection, these transduced J774.16 cells were harvested and seeded onto a six-well plate with supplemented DMEM containing puromycin at 5 µg/mL.

After reaching confluence, the cells were harvested, and total RNA was extracted using Trizol® (Thermo Fisher). RNA was analyzed by electrophoresis with 1% agarose, gene expression evaluated by quantitative polymerase chain reaction (qPCR) using SYBR® Green to generate the amplification signal, and the fold-changes were calculated using the $2^{-\Delta\Delta Ct}$ method.

*Fungal killing assay* - For CFU experiments, stably transduced J774.16 cells were seeded onto a 96-well plate at $2 \times 10^4$ cells per well in supplemented DMEM and activated with murine interferon (IFN)-$\gamma$ at 200 U/mL plus LPS at 1 µg/mL. After 24 h of activation and adhesion, the J774.16 cells were co-incubated with *P. brasiliensis* suspensions. To prepare these fungal suspensions, *P. brasiliensis* yeast cells were scraped from solid media and suspended in phosphate buffered saline solution (PBS). After vortexing with 2 - 4 mm glass beads for 30 s, large clumps were removed by decanting and the suspension strained through a 40 µm cell strainer. The viable cell density on the resulting suspension was counted in a hemocytometer using the vital dye Phloxine B. J774.16 cells were co-incubated with *P. brasiliensis* for 24 h, with a multiplicity of infection (MOI) of one. After this period, macrophages were lysed with sterile distilled water and fungal CFUs were counted by plating the same dilution for each well onto BHI agar plates and incubating at 37ºC until colonies appeared (five to seven days). Controls included untransduced J774.16 cells and wells with no macrophages. The experiment was repeated independently three times in different days, each with four or five wells per condition.

*Co-incubation of macrophages and Paracoccidioides spp. for LC3 immunofluorescence* - In different experiments, macrophages were either plated onto glass-bottom dishes (Mattek®) or on 24-well plates with sterile circular coverslips for 24 h. *P. brasiliensis* yeast cells were harvested from five-day old culture plates by scraping the surface of the fungal mat, vortexing the cells in PBS, passing the suspension through a 40 µm cell strainer and measuring cell density in a hemocytometer. Fungal cells were inoculated onto the plated macrophages at a MOI of one. The dishes were incubated for 12 to 24 h at 37ºC in the presence of 5% $CO_2$ to allow infection. Afterwards, the plates were processed for immunofluorescence as described below. In some experiments, we added the Syk-selective tyrosine kinase inhibitor piceatannol (PIC) at 30 µM (Invivogen, San Diego, CA, catalog # tlrl-pct) or the nicotinamide adenine dinucleotide phosphate oxidase (NADPH) inhibitor diphenyleneiodonium chloride (DPI) at 20 µM (Sigma-Aldrich, Sant Louis, Missouri, catalog #D2926) to the dishes. Inhibitors were added 10 min before stimulation and remained in culture for the duration of the experiments. To minimize potential bias in the quantification of LC3 recruitment, all scoring was performed by a trained technician blinded to the experimental groups. Additionally, *C. albicans* infection was included as a positive control for LAP induction, confirming the reliability of the scoring methodology.

*LC3 immunolocalization* - After 12 or 24 h of infection, the cells were fixed with ice-cold methanol for 10 min and washed with PBS. After that, they were incubated with a 1% BSA solution in PBS containing primary antibody (rabbit polyclonal IgG against human LC3, 1:1000 dilution, Santa Cruz Biotechnology) for 1 h at 37ºC. They were then washed three times with PBS and incubated with the secondary antibody (goat IgG against rabbit IgG conjugated with AlexaFluor® 488, Thermo Fisher Scientific) diluted 1:2000 in the same conditions as the primary one. Calcofluor White at 1 g/L was used in some experiments to label the fungal cell wall. After straining, the cells were washed three times with PBS and the glass-bottom dishes (or coverslips) were mounted with ProLong Gold Antifade Mountant (Thermo Fisher Scientific). We included two negative controls [Supplementary data (Fig. 1A)]: (1) unstained macrophages and *Paracoccidioides* cells (autofluorescence); (2) co-cultures of macrophage and *Paracoccidioides* cells with the secondary but without the primary antibody (non-specific binding of the secondary antibody). Additional experiments with no macrophages, only *Paracoccidioides* spp. cells, showed that the LC3 antibody does not bind to the fungal cell [Supplementary data (Fig. 1B)]. Samples were documented in a Zeiss Axio Observer Z1 epifluorescence microscope equipped with a 63x NA 1.4 oil immersion objective and a cooled CCD camera. Image stacks were deconvolved with a constrained iterative algorithm on Zeiss ZEN and then processed on ImageJ and Adobe Photoshop. No non-linear modifications were made to the images.

*Statistical analysis* - For *ATG5* knockdown, analysis of variance (ANOVA) and Dunnett's multiple comparison pos-hoc tests were performed on Graphpad Prism. For LAP quantitative analysis, a Fisher's exact test was performed to compare proportions of fungi on LC3-positive vacuoles on Graphpad Prism. For CFU analysis, a mixed-analysis ANOVA was used, with shRNA as a fixed effect and replicate as a random factor. Pairwise comparisons were made using Tukey's HSD to correct for multiple comparisons. This CFU analysis was performed in R using the MultComp package.[29]

## RESULTS

LC3 is recruited to phagosomes containing Paracoccidioides spp. in murine macrophages - To investigate whether macrophages use LAP against *Paracoccidioides* spp., we performed immunofluorescence microscopy. LC3 was detected in phagosomes containing either *P. brasiliensis* or *P. lutzii* across multiple macrophage types (RAW264.7, J774.16, and BMM) after 12 or 24 h of co-incubation (Fig. 1A-C). Controls for autofluorescence and non-specific secondary antibody binding were negative [Supplementary data (Fig. 1)]. Using calcofluor white, we could see that LC3 accumulates around the fungal cell wall (Fig. 2).

Interestingly, LC3 localization was more frequently observed around fungal daughter cells compared to mother cells (Fig. 1B). Additionally, we detected LC3

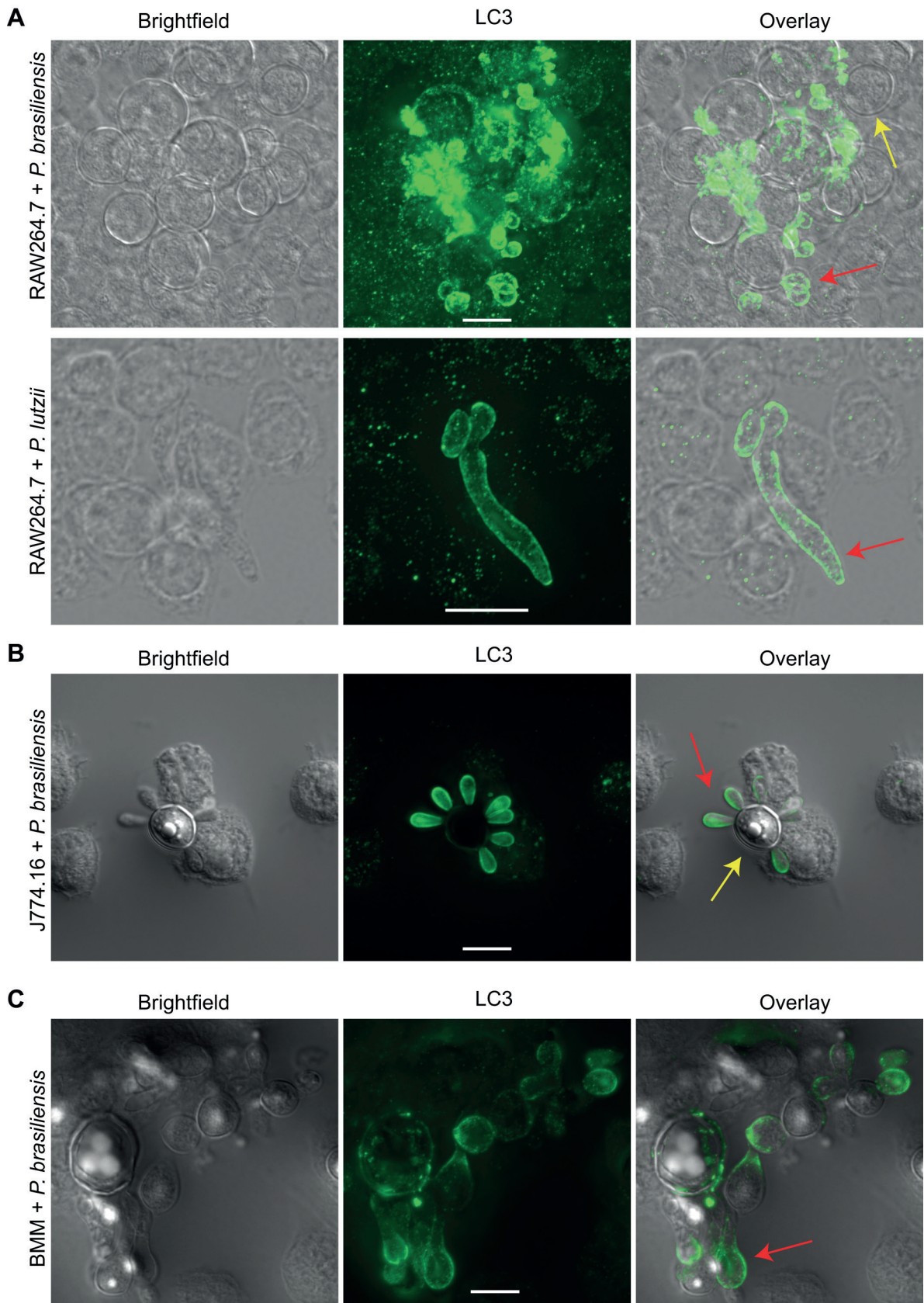

Fig. 1: LC3 associated phagocytosis (LAP) is activated in murine macrophages against *Paracoccidioides* spp. (A) LAP was detected after 24 h in RAW264.7 incubated with *Paracoccidioides brasiliensis* and *P. lutzii*. (B) The same phenomenon was confirmed after 12 h of the *P. brasiliensis* interaction with J774.16 and (C) bone marrow derived macrophages (BMM). Experiments were repeated at least twice on different days and had similar results. The red arrows indicate fungi that are surrounded by LC3, whereas the yellow arrows point to fungi that are not. Scale bars: 10 μm.

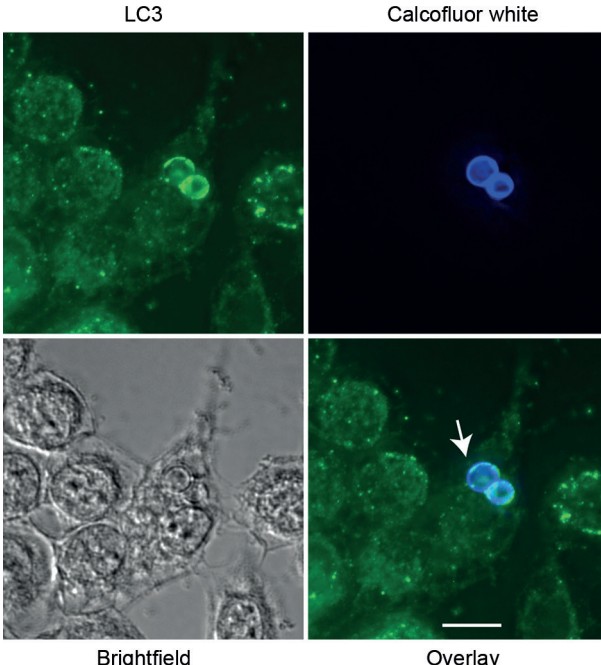

LC3

Calcofluor white

Brightfield

Overlay

Fig. 2: accumulation and colocalization of LC3 in the *Paracoccidioides* cell wall. *Paracoccidioides brasiliensis* cells were dyed with calcofluor white before their interaction with J774.16 macrophages. LC3 associated phagocytosis (LAP) was detected after 24 h of interaction, and the signal could be observed around the fungal cell wall (arrow). Scale bar: 10 μm.

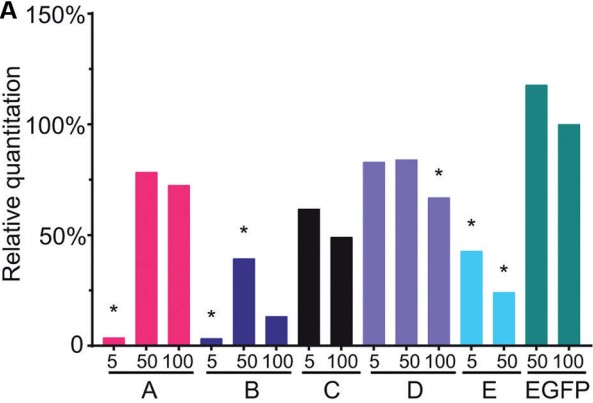

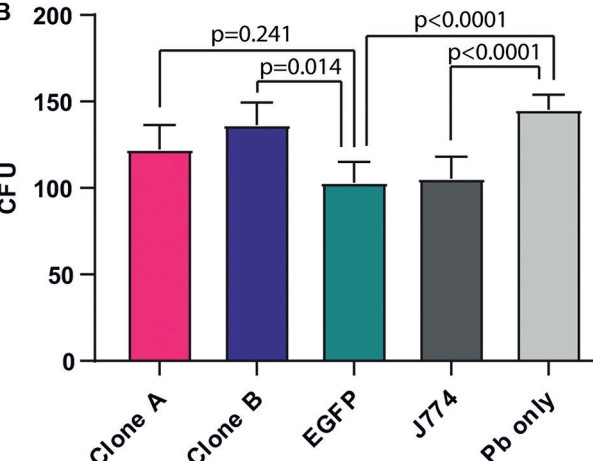

Fig. 3: ATG5 knockdown efficiency and impact on fungal burden in *Paracoccidioides*-infected macrophages. (A) Relative quantification of ATG5 expression in J774.16 macrophages transduced with five different lentiviral shRNA constructs that target ATG5 (clones A-E) compared to a negative control vector that targets a gene that is not present on the cells (EGFP). Bars represent ATG5 mRNA levels at three different multiplicities of infection (MOI: 5, 50, 100). Clones A and B showed the strongest knockdown efficiency (asterisks indicate p < 0.05 compared to the EGFP negative control). (B) Fungal burden [colony forming unit (CFU) counts] in *Paracoccidioides brasiliensis* (Pb)-infected macrophages at 48 h post-infection. ATG5-silenced clones A and B are compared to negative controls including untransduced macrophages (J774) and cells in which the shRNA target is not present (EGFP). "Pb only" indicates fungal growth in the absence of macrophages. Data represent mean ± standard deviation (SD) from three independent experiments. Statistical analysis was performed using one-way analysis of variance (ANOVA) with Tukey's HSD multiple comparison post-hoc test; p-values are shown above the bars.

around *Paracoccidioides* spp. cells that were clearly outside macrophages. A final striking observation is that LC3 recruitment was not universal, as only a subset of phagosomes displayed LC3. These findings demonstrate that LAP is activated in macrophages during interactions with *Paracoccidioides* spp., albeit in a selective manner.

LAP is important in the murine macrophage response to P. brasiliensis - To explore the functional significance of LC3 recruitment, we conducted loss-of-function experiments targeting ATG5, a gene essential for autophagy and LAP.[26] J774.16 macrophages were transduced with lentiviral vectors encoding shRNAs against ATG5. Two shRNAs (clones A and B) achieved significant knockdown, reducing ATG5 expression by approximately 97% compared to controls (Fig. 3A). A significant reduction in antifungal activity in comparison with the non-silenced control was observed in ATG5-silenced clone B, but not on clone A. Targeting a sequence that does not exist on the macrophage, EGFP, had no effect, confirming that the reduced antifungal activity is specific for ATG5 silencing in clone B (Fig. 3B). These findings indicate that LAP contributes to macrophage antifungal responses against *Paracoccidioides* spp.

Distinct mechanisms of LAP activation - To determine whether LAP activation against *Paracoccidioides* spp. involves similar pathways to those used against other fungi, we used pharmacological inhibitors targeting Syk kinase and the NADPH oxidase, which are required for LAP induction in macrophages infected with C. albicans[21] and A. fumigatus.[17] Syk inhibition by piceatannol increased LC3 recruitment to phagosomes containing P. brasiliensis, while NADPH oxidase inhibition with DPI had no effect (Fig. 4 and Table).

As these results differ from what was observed with other fungi, we also repeated the experiments with C. albicans. As shown on Supplementary data (Fig. 2) and Table, the replication of previous studies[21] confirms our results.

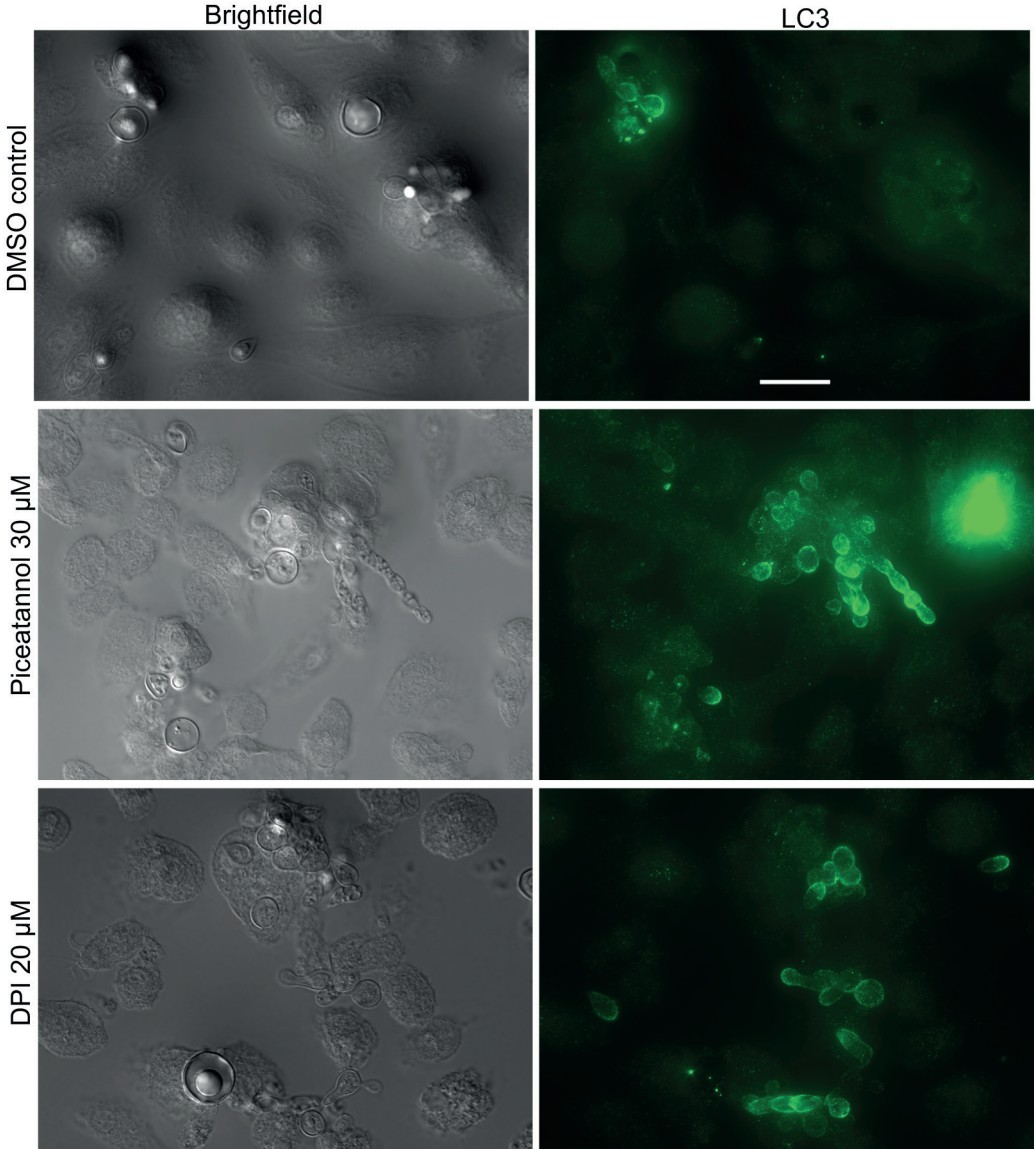

Fig. 4: effect of Syk and nicotinamide adenine dinucleotide phosphate oxidase (NADPH) inhibitor on bone marrow derived macrophages (BMM) LC3 associated phagocytosis (LAP) against *Paracoccidioides brasiliensis*. Piceatannol (PIC) at 30 μM) and diphenyleneiodonium chloride (DPI) at 20 μM were used to inhibit Syk and NADPH oxidase, respectively. Syk inhibition led to an increase in LAP, whereas NADPH oxidase inhibition did not seem to affect LAP. White arrows indicate the dotted pattern of LC3 protein inside macrophages (not associated with phagocytosed fungi). Scale bar: 20 μm.

## DISCUSSION

Our findings demonstrate that macrophages deploy LAP as part of their response to *Paracoccidioides* spp. Using immunofluorescence, we observed LC3 recruitment to phagosomes containing different *Paracoccidioides* spp. across multiple macrophage types, including RAW264.7, J774.16, and bone marrow-derived macrophages. Interestingly, this recruitment was selective, occurring in only a subset of phagosomes, with LC3 frequently observed around fungal daughter cells but not mother cells. This pattern may reflect differences in the composition of the fungal cell wall, as surface microbial molecular patterns are key LAP triggers.[22] Additionally, LC3 was detected around apparently extracellular fungal cells, a phenomenon that had been previously observed with *C. neoformans*.[18] This could represent non-lytic exocytosis (vomocytosis) of the phagosomal contents, a process previously reported in interactions between *Paracoccidioides* spp. and amoebae;[30] testing this speculation, however, is beyond the scope of the present work.

The functional significance of this LC3 recruitment was confirmed through loss-of-function experiments targeting ATG5, a gene that is essential for autophagy and LAP. ATG5 knockdown significantly reduced macrophage antifungal activity, as evidenced by increased fungal viability in CFU assays. These results support the role of LAP in macrophage-mediated control of *P. brasiliensis*. However, the modest reduction in an-

TABLE

Recruitment of LC3 to vacuoles containing *Paracoccidioides brasiliensis* and *Candida albicans* in primary macrophages. Bone marrow derived macrophages (BMMs) were infected with either fungi in the presence or absence of Syk and nicotinamide adenine dinucleotide phosphate oxidase (NADPH) inhibitors and processed for immunofluorescence microscopy as shown in Fig. 4. Individual macrophages with at least one phagocytosed yeast were then observed and scored based on the presence or absence of LC3 surrounding the internalized fungi

| Fungus | Treatment | Macrophages[a] | LAP[b] | Percentage[c] | Odds ratio | p-value[d] |
|---|---|---|---|---|---|---|
| *P. brasiliensis* | DMSO | 218 | 39 | 17.9% | | |
| | 30 µM PIC | 208 | 62 | 29.8% | 1.666 | 0.0267 |
| | 20 µM DPI | 222 | 47 | 21.2% | 1.183 | 0.4822 |
| *C. albicans* | DMSO | 103 | 58 | 56.3% | | |
| | 30 µM PIC | 75 | 30 | 40.0% | 0.71 | 0.2314 |
| | 20 µM DPI | 119 | 25 | 21.0% | 0.373 | 0.0003 |

*a*: number of macrophages that were observed to have phagocytosed at least one *P. brasiliensis* cell; *b*: number of macrophages among those in which the phagocytosed fungi were surrounded by LC3; *c*: percentage of macrophages in which the phagocytosed fungi were surrounded by LC3; *d*: Fisher's exact test; LAP: LC3 associated phagocytosis; DMSO: dimethyl sulfoxide; PIC: piceatannol; DPI: diphenyleneiodonium chloride.

tifungal activity observed, even in ATG5-silenced macrophages, suggests that LAP operates as part of a broader immune response rather than acting as the sole antifungal process.

The CFU results align with previous work from our group on macrophages and dendritic cells from susceptible and resistant mouse strains,[31] an indication that LAP could play a role in *in vivo* responses to *Paracoccidioides* spp. However, care is warranted in reaching strong conclusions in this regard because on macrophages infected with the closely related *H. capsulatum*, LAP is actually detrimental to the host and exploited by the fungus to survive.[23] The literature on antifungal LAP in macrophages is rife with apparent contradictions that highlight how complex this mechanism is. In macrophages infected with *C. neoformans in vitro*, for instance, we found that LAP was host-protective[18] but another group found it benefited the pathogen[19,32] In invasive candidiasis models, we[18] and others[17,33,34] found that autophagy was host-protective, whereas other experiments concluded it was not necessary for proper responses to *C. albicans*.[35]

Mechanistically, our results suggest distinct pathways of LAP activation in macrophages infected with *Paracoccidioides* spp. compared to other fungi such as *C. albicans*,[36] *A. fumigatus*[17] or *Histoplasma capsulatum*.[37] Pharmacological inhibition of Syk kinase, which is essential for LAP activation in *C. albicans*, unexpectedly increased LC3 recruitment to phagosomes containing *P. brasiliensis*. Similarly, the inhibition of NADPH oxidase, another critical pathway for LAP activation in other fungal infections, had no effect on LAP in *P. brasiliensis*. The divergent effects of these inhibitors suggest that macrophages rely on distinct signaling pathways to activate LAP in response to *Paracoccidioides* spp. Differences in fungal cell wall composition, particularly in exposure of α- and β-glucans and other pattern recognition receptor ligands,[38,39] may underlie these mechanistic differences.

Despite the strength of these findings, several limitations should be considered. The *in vitro* nature of our experiments restricts their applicability to the complex immune environment of *in vivo* infections. Additionally, the limited antifungal activity observed in macrophage cultures may reflect intrinsic limitations of the J774.16 cell line. Future studies should investigate LAP's role in murine PCM models, including its interaction with other immune pathways, such as cytokine signaling and adaptive immune responses.

Nevertheless, this activation of LAP against *Paracoccidioides* spp. has potential therapeutic implications. Targeting host-directed pathways with pharmacological activators of LAP or compounds that enhance downstream effector functions to enhance macrophage antifungal activity could be explored as adjunctive PCM therapies. This would reduce reliance on current antifungal therapies, which are often associated with toxicity, long treatment, and the risk of relapse;[40,41,42,43] and possibly lead to better outcomes. A new generation of specific autophagy-modulating compounds[35,44,45,46] may hold promise in this regard.

In summary, our findings add to the growing body of scientific evidence that the intracellular events after the uptake of fungal cells by phagocytes are protean and more research is needed to generate a consistent picture of these phenomena. Nevertheless, they provide new insights into antifungal immunity and underscore the potential of LAP as a target for host-directed therapies to combat systemic mycoses such as PCM.

### ACKNOWLEDGEMENTS

To John Reidhaar-Olson, from the Albert Einstein College of Medicine shRNA Core Facility, for the assistance with shRNA experiments.

### AUTHORS' CONTRIBUTION

Conceptualization - AMN, AC and PA; writing-original draft preparation - GPOJ, HRS, KCMG, PA and AMN; supervision - AMN, ISP and MSSF; methodology - HCP, ISP,

PA, KCMG, TKSB, KTR, SF, FCKG, LFF, ARN, FAH, GPOJ and HRS; formal analysis - GPOJ, HRS, SF and AMN; writing - review & editing - HCP, ISP, PA, AC and MSSF; funding acquisition - AMN, ISP, AC and MSSF. All authors have read and agreed to the published version of the manuscript. The authors declare that they have no conflicts of interest.

## DATA AVAILABILITY

The contents underlying the research text are included in the manuscript.

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

# OPEN PEER REVIEW

Memórias do IOC thanks the anonymous reviewers for their contribution to the peer review of this work.

**FIRST REVIEW ROUND**

REVIEWERS' COMMENTS

**REVIEWER #1**

In the manuscript by Oliveira Junior et al., entitled "LC3-associated phagocytosis in macrophage responses to Paracoccidioides spp" the authors investigated the role of LC3- associated phagocytosis (LAP) in the context of Paracoccidioidomycosis spp. infection. First of all, the authors showed by immunofluorescence that LC3 associated phagocytosis is activated in murine macrophages, utilizing both cells lines and primary macrophages. Interestingly, some LC3 detected were in extracellular fungus. To explore the functional relevance of LAP, the authors silenced ATG5 using shRNA and observed an increased fungal burden in infected macrophages, suggesting that ATG5 is important for LAP in Paracoccidioides infection. Furthermore, they inhibited components previously implicated in LAP during other fungal infections and found these were not required in this context.

The obersvation that LAP is active during Paracoccidioidomycosis is intriguing and of broad interest to the field. Future studies clarifying if LAP is beneficial or detrimental in murine Paracoccidioidomycosis will help us better understand its role in this context. The manuscript is well written but I have some concerns about the methodology and some minor improvements that should be addressed to strengthen the manuscript.

Major concerns:

1 – The authors utilized 20 ng/mL of GM-CSF to generate macrophages, although GM-CSF is known to generate these immune cells, other cells such as dendritic cells are found as well (PMID: 26084029 and PMID: 27788572). It is important that the authors clarify whether they performed any purification after cell generation of they utilized the cell pool.

2 - Another concern regarding the generation of macrophages is that the authors utilized non-adherent plates, but these macrophages are well-known for being adherent, I wonder if the authors had a reason for non-treated plates or if they checked cell viability/adherence/differentiation (PMID: 26084029 and PMID: 27788572).

Minor concerns:

1 – Why authors used anti-human LC3 if murine macrophages were being used?

2 – Figure 3 caption could be improved.

a. There is no mention of what A, B, C, D, and E stands for.

b. EGFP controls should be described to the methods section as well.

c. It should be explicit in 3B that J774 is the non-silenced control cells.

3 – In lines 237-239 it should say that the reduced antifungal activity was only achieved in clone B, since clone A had no significant difference.

4 – In line 264 the authors suggest that vomocytosis might occur. If possible, providing representative images or videos would strengthen this claim.

5 – In line 271 the authors say there is a modest reduction in fungal burden, but that is the opposite as the figure shows.

6 – In line 301 a reference should be added to support the statement regarding β-glucan.

**REVIEWER #2**

The manuscript by Oliveira Junior et al. presents interesting and relevant findings on the role of LC3-associated phagocytosis (LAP) in the macrophage response to Paracoccidioides spp., contributing to our understanding of host-pathogen interactions in paracoccidioidomycosis (PCM). The demonstration of LC3 recruitment to phagosomes and the use of RNA interference to highlight the importance of ATG5 are strengths of the study. Moreover, the observation that canonical LAP signaling pathways (Syk and NADPH oxidase) are not required in this context adds novelty and suggests distinct mechanisms of fungal recognition.

However, the manuscript requires clarification and revision in several areas before it can be considered for publication.

Major Concerns

1. The manuscript relies heavily on immunofluorescence analysis. However, many of the presented images do not clearly distinguish between phagocytosed fungi and those merely adhering to macrophage surfaces. The microscopy system used may not be optimal for resolving this distinction. In Figure 1, it is unclear what the authors are highlighting. The LC3 staining is  macrophage-associated, yet based on the presented images, there seems to be LC3

signal external to the phagocytic cells. In some images (e.g., Fig. 1B and 1C), fungal structures are labeled both on the macrophage-interacting and non-interacting sides. A possible solution would be to stain the actin cytoskeleton of macrophages with phalloidin to more clearly delineate cell boundaries and confirm intracellular localization.

2. An important methodological concern that should be addressed is whether the anti-LC3 antibody used in the study might cross-react with LC3 homologs in Paracoccidioides spp. Could the observed labeling correspond, at least in part, to fungal LC3-like proteins? The authors should discuss this possibility and ideally demonstrate that the LC3 signal is indeed specific to host LC3 and not a fungal ortholog.

3. Similar concerns arise with the data shown in Figure 4 using pharmacological inhibitors of Syk and NADPH oxidase. The images do not provide convincing evidence regarding the precise localization of LC3 staining and whether the signal corresponds to phagocytosed fungi or extracellular particles.

4. Interpretation of Figure 3B. The authors conclude that ATG5 knockdown leads to reduced macrophage antifungal activity. However, an alternative explanation, enhanced fungal uptake, cannot be ruled out based on the current data. To support their interpretation, the authors should present phagocytic index data under the same conditions used in this assay to clarify whether the difference in CFU reflects altered killing or increased phagocytosis.

5. Discussion. The hypothesis that differences in the fungal cell wall account for variations in LAP induction is intriguing but underdeveloped. The authors should support this claim with references to existing literature and, if possible, provide data on basic fungal surface features under the conditions used.

This is a promising and potentially impactful study. However, to reach its full potential, the manuscript requires revisions to enhance methodological clarity, support key conclusions, and improve figure interpretation. I encourage the authors to address the issues above in a revised version.

Minor Points

a. The multiplicity of infection (MOI) used in the phagocytosis assays is not specified and should be clearly stated.

b. Figure legends are insufficiently descriptive and occasionally include interpretations of results rather than objective descriptions. These should be revised for clarity and accuracy.

## AUTHORS' RESPONSE TO THE REVIEWERS

Dear Dr. Brandão,

Thank you very much for considering our manuscript, and for sending us the reviewer comments. We would also like to thank the reviewers, whose thoughtful comments helped us improve a lot our manuscript. We are confident our manuscript is now on a format that is ready for publication. I will paste below the reviewer comments, along with answers to each of them.

Sincerely,
Andre

REVIEWER COMMENTS:

Reviewer: 1

Reviewer comments: In the manuscript by Oliveira Junior et al., entitled "LC3-associated phagocytosis in macrophage responses to Paracoccidioides spp" the authors investigated the role of LC3- associated phagocytosis (LAP) in the context of Paracoccidioidomycosis spp. infection. First of all, the authors showed by immunofluorescence that LC3 associated phagocytosis is activated in murine macrophages, utilizing both cells lines and primary macrophages. Interestingly, some LC3 detected were in extracellular fungus. To explore the functional relevance of LAP, the authors silenced ATG5 using shRNA and observed an increased fungal burden in infected macrophages, suggesting that ATG5 is important for LAP in Paracoccidioides infection. Furthermore, they inhibited components previously implicated in LAP during other fungal infections and found these were not required in this context.

The obersvation that LAP is active during Paracoccidioidomycosis is intriguing and of broad interest to the field. Future studies clarifying if LAP is beneficial or detrimental in murine Paracoccidioidomycosis will help us better understand its role in this context. The manuscript is well written but I have some concerns about the methodology and some minor improvements that should be addressed to strengthen the manuscript.

Response: Thank you very much for the positive comments on our manuscript.

Major concerns:

1 – The authors utilized 20 ng/mL of GM-CSF to generate macrophages, although GM-CSF is known to generate these immune cells, other cells such as dendritic cells are found as well (PMID: 26084029 and PMID: 27788572). It is important that the authors clarify whether they performed any purification after cell generation of they utilized the cell pool.

Response: We thank the reviewer for bringing up this important point. GM-CSF stimulation of bone marrow progenitors indeed generates a heterogeneous population. Removing non-adherent cells does decrease the heterogeneity, as pointed in one of the publications the reviewer mentioned (PMID 27788572): around 80% of the adherent cells in this GM-CSF concentration are bone marrow macrophages.

Action: We have clarified this point explicitly in the Methods section by adding the sentence: "a condition that results in cultures with approximately 80% macrophages (Siqueira, Fraga et al. 2016). [lines 110 - 111]

2 - Another concern regarding the generation of macrophages is that the authors utilized non-adherent plates, but these macrophages are well-known for being adherent, I wonder if the authors had a reason for non-treated plates or if they checked cell viability/adherence/differentiation (PMID: 26084029 and PMID: 27788572).

Response: We used plates that had not been vacuum plasma treated, but this does not mean that these plates are non-adherent. To the contrary, macrophages do adhere firmly to these plates and have to be enzymatically detached, whereas dendritic cells typically stay in suspension or loosely attached. The use of non-treated culture plates is a well-established approach in GM-CSF differentiation protocols. Cell viability was consistently above 95% as assessed by trypan blue exclusion, and the differentiated macrophages exhibited typical morphology with large cytoplasm and prominent pseudopodia.

Action: We added the following phrase to the Methods section: "Cell viability was consistently >95% as measured by trypan blue exclusion." [lines 118 - 119]

Minor concerns:
1 – Why authors used anti-human LC3 if murine macrophages were being used?

Response: The manufacturer's datasheet states that this antibody has been validated for detection of LC3 proteins from mouse, rat, and human origin. This is not unexpected, given how conserved LC3 is in both species. The immunofluorescence pattern we observed is consistent with that.

Action: None.

2 – Figure 3 caption could be improved.
a. There is no mention of what A, B, C, D, and E stands for.

Response: We agree the figure caption was not clear.

Response: We have modified the figure caption to explain what the labels mean [lines 354 - 367].

b. EGFP controls should be described to the methods section as well.

Response: We agree a better description is necessary not only in the methods section, but in the rest of the manuscript as well.

Action: We have added to the Methods section the following sentence: "As negative control, we used a similar vector encoding an shRNA that targets the enhanced green fluorescent protein (EGFP) gene, a sequence that is not present in the cell." [lines 123 - 125].

We have also improved the description of these controls on the Results section Results [lines 235 – 237] and Figure 3 legend [lines 357 – 358].

c. It should be explicit in 3B that J774 is the non-silenced control cells.

Response: We agree.

Action: As described above, we have thoroughly improved the legend to figure 3. The sentence we added to address this reads: "ATG5-silenced clones A and B are compared to negative controls, including untransduced macrophages (J774) and cells in which the shRNA target is not present (EGFP)" [lines 362 - 364].

3 – In lines 237-239 it should say that the reduced antifungal activity was only achieved in clone B, since clone A had no significant difference.

Response: We agree.

Action: We have reformulated that section of the text to read: "A significant reduction in antifungal activity in comparison with the non-silenced control was observed in ATG5-silenced clone B, but not on clone A. Targeting a sequence that does not exist on the macrophage, EGFP, had not effect, confirming that the reduced antifungal activity is specific for ATG5 silencing in clone B (Figure 3B)." [lines 233 - 238].

4 – In line 264 the authors suggest that vomocytosis might occur. If possible, providing representative images or videos would strengthen this claim.

Response: We agree it would be great to have videomicroscopy results to measure non-lytic exocytosis (vomocytosis). We did just that on a previous work with Cryptococcus neoformans (Nicola et al., 2012), but that required a specialized spinning disk confocal microscope and two years of work. However interesting that subject is, it is beyond the scope of this work. We chose to keep this speculation on the discussion section, but added a sentence to clarify that it is just speculation.

Action: We added the following sentence to the Discussion: "testing this tantalizing speculation, however, is beyond the scope of the present work." [lines 266 - 267].

5 – In line 271 the authors say there is a modest reduction in fungal burden, but that is the opposite as the figure shows.

Response: We thank the reviewer for picking up this mistake.

Action: We have corrected the text to accurately reflect our findings: "However, the modest reduction in an-

tifungal activity observed, even in ATG5-silenced macrophages, suggests that LAP operates as part of a broader immune response rather than acting as the sole antifungal process." [Line 272].

6 – In line 301 a reference should be added to support the statement regarding β-glucan.
Response: We agree.
Action: We have added two references to that sentence, which now reads: "Differences in fungal cell wall composition, particularly in surface exposure of α- and β-glucans and other pattern recognition receptor ligands (San-Blas, San-Blas et al. 1977, Puccia, Vallejo et al. 2011), may underlie these mechanistic differences." [Lines 297 - 300].

Reviewer: 2
Reviewer comments: The manuscript by Oliveira Junior et al. presents interesting and relevant findings on the role of LC3-associated phagocytosis (LAP) in the macrophage response to Paracoccidioides spp., contributing to our understanding of host-pathogen interactions in paracoccidioidomycosis (PCM). The demonstration of LC3 recruitment to phagosomes and the use of RNA interference to highlight the importance of ATG5 are strengths of the study. Moreover, the observation that canonical LAP signaling pathways (Syk and NADPH oxidase) are not required in this context adds novelty and suggests distinct mechanisms of fungal recognition.
We thank the reviewer for pointing out our work's strength.
However, the manuscript requires clarification and revision in several areas before it can be considered for publication.

Major Concerns
1. The manuscript relies heavily on immunofluorescence analysis. However, many of the presented images do not clearly distinguish between phagocytosed fungi and those merely adhering to macrophage surfaces. The microscopy system used may not be optimal for resolving this distinction. In Figure 1, it is unclear what the authors are highlighting. The LC3 staining is macrophage-associated, yet based on the presented images, there seems to be LC3 signal external to the phagocytic cells. In some images (e.g., Fig. 1B and 1C), fungal structures are labeled both on the macrophage-interacting and non-interacting sides. A possible solution would be to stain the actin cytoskeleton of macrophages with phalloidin to more clearly delineate cell boundaries and confirm intracellular localization.
Response: We share the reviewer's wonders in this case, and have done so since our first LC3 immunofluorescence experiments with other fungi (Cryptococcus neoformans and Candida albicans) almost 20 years ago. In all of them, we have observed similar LC3 recruitment to the surface of fungi that are clearly inside macrophages, but also to fungi that appear to be outside any phagocyte. This happened reproducibly in tests done by several researchers, in two different labs in different countries. This puzzling observation is not a technical issue, as high aperture oil immersion objectives and constrained iterative deconvolution allow optical sectioning at the very limit of resolution in conventional fluorescence microscopy. Thus, repeating the experiment with fluorescent phalloidin would bring no additional insights.
We have speculated in the present manuscript that this could be a result of non-lytic exocytosis, something that we have shown directly in our previous work with other fungi (Nicola et al., 2012). As explained to the first reviewer, however, testing so takes years and is the subject of future work, not this manuscript.
Action: We have modified the following sentence in the discussion to clarify this issue:
"This could represent non-lytic exocytosis (vomocytosis) of the phagosomal contents, a process previously reported in interactions between Paracoccidioides spp. and amoebae (Gonçalves, Ferreira et al. 2019); testing this speculation, however, is beyond the scope of the present work." [Lines 264-267]

2. An important methodological concern that should be addressed is whether the anti-LC3 antibody used in the study might cross-react with LC3 homologs in Paracoccidioides spp. Could the observed labeling correspond, at least in part, to fungal LC3-like proteins? The authors should discuss this possibility and ideally demonstrate that the LC3 signal is indeed specific to host LC3 and not a fungal ortholog.
Response: We agree with the reviewer. Using exactly the same staining protocol but with Paracoccidioides spp. cells results in no fluorescence at all (see new Figure S1B). This is expected, as the fungal LC3 homolog (Atg8) is very different from the mammalian counterpart that was used to raise the commercial antibodies we used.
Action: We added to Figure S1 an additional panel, and also added to the Materials and Methods section the following sentence: "Additional experiments with no macrophages, only Paracoccidioides spp. cells, showed that the LC3 antibody does not bind to the fungal cell (Figure S1B)." [Lines 198 – 199].

3. Similar concerns arise with the data shown in Figure 4 using pharmacological inhibitors of Syk and NADPH oxidase. The images do not provide convincing evidence regarding the precise localization of LC3 staining and whether the signal corresponds to phagocytosed fungi or extracellular particles.
Response: We agree with the reviewer regarding the issue of differentiating if the fungi surrounded by LC3 are inside or outside the macrophages, as discussed above. We respectfully disagree with the statement that the images are not convincing evidence of precise localization of LC3 staining. We did see very reproducible localization of LC3 around fungal cells. This is the same pattern that we and others have observed before on multiple fungal species that were co-incubated with macrophages.
Action: None.

4. Interpretation of Figure 3B. The authors conclude that ATG5 knockdown leads to reduced macrophage anti-fungal activity. However, an alternative explanation, enhanced fungal uptake, cannot be ruled out based on the current data. To support their interpretation, the authors should present phagocytic index data under the same conditions used in this assay to clarify whether the difference in CFU reflects altered killing or increased phagocytosis.

Response: We appreciate this thoughtful comment. It is important to note, though, that as we described in the materials and methods, at no moment between adding fungi and collecting the well contents did we wash the wells. This means that higher phagocytosis does not explain the observed difference in CFU, as all fungi that were added to each well remained there until collected after 24 h.

Action: None.

5. Discussion. The hypothesis that differences in the fungal cell wall account for variations in LAP induction is intriguing but underdeveloped. The authors should support this claim with references to existing literature and, if possible, provide data on basic fungal surface features under the conditions used.

Response: We are glad the reviewer found this intriguing, we also did. However, we have not pursued this avenue further right now. Although it certainly will be the foundation for future experiments, we argue that our present manuscript, a first demonstration that LC3-associated phagocytosis plays a role in the interaction of macrophages and Paracoccidioides spp, is a full story as is. We agree with the reviewer's suggestion to support this speculation with more references, so we have added two references about the Paracoccidioides cell wall and its effects on virulence. We did not, however, develop this any further because we have not experimentally addressed the issue.

Action: As explained in response to the first reviewer, we have updated a sentence in the discussion to: "Differences in fungal cell wall composition, particularly in surface exposure of α- and β-glucans and other pattern recognition receptor ligands (San-Blas, San-Blas et al. 1977, Puccia, Vallejo et al. 2011), may underlie these mechanistic differences." [Lines 297 - 300].

This is a promising and potentially impactful study. However, to reach its full potential, the manuscript requires revisions to enhance methodological clarity, support key conclusions, and improve figure interpretation. I encourage the authors to address the issues above in a revised version.

Minor Points
a. The multiplicity of infection (MOI) used in the phagocytosis assays is not specified and should be clearly stated.
Response: We thank the reviewer for noting this omission.
Action: We have updated the Materials and Methods section to clarify we used MOI of 1 in all experiments [lines 161 - 162].
b. Figure legends are insufficiently descriptive and occasionally include interpretations of results rather than objective descriptions. These should be revised for clarity and accuracy.
Response: We agree.
Action: We have carefully revised the figure legends.

## SECOND REVIEW ROUND

REVIEWERS' COMMENTS

### REVIEWER #1

I thank the authors for addressing all of my comments. The manuscript has improved and I have no further concerns.

### REVIEWER #2

The authors have satisfactorily addressed all the concerns raised in my previous review. The revised version of the manuscript adequately incorporates the suggested clarifications and improvements. I have no further comments.

