## [Reviewer Report · FIRST REVIEW ROUND - REVIEWERS COMMENTS]

## REVIEWER #1

In the manuscript by Oliveira Junior et al., entitled “LC3-associated phagocytosis in macrophage responses to Paracoccidioides spp” the authors investigated the role of LC3- associated phagocytosis (LAP) in the context of Paracoccidioidomycosis spp. infection. First of all, the authors showed by immunofluorescence that LC3 associated phagocytosis is activated in murine macrophages, utilizing both cells lines and primary macrophages. Interestingly, some LC3 detected were in extracellular fungus. To explore the functional relevance of LAP, the authors silenced ATG5 using shRNA and observed an increased fungal burden in infected macrophages, suggesting that ATG5 is important for LAP in Paracoccidioides infection. Furthermore, they inhibited components previously implicated in LAP during other fungal infections and found these were not required in this context.

The obersvation that LAP is active during Paracoccidioidomycosis is intriguing and of broad interest to the field. Future studies clarifying if LAP is beneficial or detrimental in murine Paracoccidioidomycosis will help us better understand its role in this context. The manuscript is well written but I have some concerns about the methodology and some minor improvements that should be addressed to strengthen the manuscript.

**Major concerns:**

1 – The authors utilized 20 ng/mL of GM-CSF to generate macrophages, although GM-CSF is known to generate these immune cells, other cells such as dendritic cells are found as well (PMID: 26084029 and PMID: 27788572). It is important that the authors clarify whether they performed any purification after cell generation of they utilized the cell pool.

2 - Another concern regarding the generation of macrophages is that the authors utilized non-adherent plates, but these macrophages are well-known for being adherent, I wonder if the authors had a reason for non-treated plates or if they checked cell viability/adherence/differentiation (PMID: 26084029 and PMID: 27788572).

**Minor concerns:**

1 – Why authors used anti-human LC3 if murine macrophages were being used?

2 – Figure 3 caption could be improved. a. There is no mention of what A, B, C, D, and E stands for. b. EGFP controls should be described to the methods section as well. c. It should be explicit in 3B that J774 is the non-silenced control cells.

3 – In lines 237-239 it should say that the reduced antifungal activity was only achieved in clone B, since clone A had no significant difference.

4 – In line 264 the authors suggest that vomocytosis might occur. If possible, providing representative images or videos would strengthen this claim.

5 – In line 271 the authors say there is a modest reduction in fungal burden, but that is the opposite as the figure shows.

6 – In line 301 a reference should be added to support the statement regarding β-glucan.

## REVIEWER #2

The manuscript by Oliveira Junior et al. presents interesting and relevant findings on the role of LC3-associated phagocytosis (LAP) in the macrophage response to Paracoccidioides spp., contributing to our understanding of host-pathogen interactions in paracoccidioidomycosis (PCM). The demonstration of LC3 recruitment to phagosomes and the use of RNA interference to highlight the importance of ATG5 are strengths of the study. Moreover, the observation that canonical LAP signaling pathways (Syk and NADPH oxidase) are not required in this context adds novelty and suggests distinct mechanisms of fungal recognition.

However, the manuscript requires clarification and revision in several areas before it can be considered for publication.

**Major Concerns**

1. The manuscript relies heavily on immunofluorescence analysis. However, many of the presented images do not clearly distinguish between phagocytosed fungi and those merely adhering to macrophage surfaces. The microscopy system used may not be optimal for resolving this distinction. In Figure 1, it is unclear what the authors are highlighting. The LC3 staining is macrophage-associated, yet based on the presented images, there seems to be LC3 signal external to the phagocytic cells. In some images (e.g., Fig. 1B and 1C), fungal structures are labeled both on the macrophage-interacting and non-interacting sides. A possible solution would be to stain the actin cytoskeleton of macrophages with phalloidin to more clearly delineate cell boundaries and confirm intracellular localization.

2. An important methodological concern that should be addressed is whether the anti-LC3 antibody used in the study might cross-react with LC3 homologs in Paracoccidioides spp. Could the observed labeling correspond, at least in part, to fungal LC3-like proteins? The authors should discuss this possibility and ideally demonstrate that the LC3 signal is indeed specific to host LC3 and not a fungal ortholog.

3. Similar concerns arise with the data shown in Figure 4 using pharmacological inhibitors of Syk and NADPH oxidase. The images do not provide convincing evidence regarding the precise localization of LC3 staining and whether the signal corresponds to phagocytosed fungi or extracellular particles.

4. Interpretation of Figure 3B. The authors conclude that ATG5 knockdown leads to reduced macrophage antifungal activity. However, an alternative explanation, enhanced fungal uptake, cannot be ruled out based on the current data. To support their interpretation, the authors should present phagocytic index data under the same conditions used in this assay to clarify whether the difference in CFU reflects altered killing or increased phagocytosis.

5. Discussion. The hypothesis that differences in the fungal cell wall account for variations in LAP induction is intriguing but underdeveloped. The authors should support this claim with references to existing literature and, if possible, provide data on basic fungal surface features under the conditions used.

**Minor Points**

a. The multiplicity of infection (MOI) used in the phagocytosis assays is not specified and should be clearly stated.

b. Figure legends are insufficiently descriptive and occasionally include interpretations of results rather than objective descriptions. These should be revised for clarity and accuracy.

## AUTHORS' RESPONSE TO THE REVIEWERS

Dear Dr. Brandão, Thank you very much for considering our manuscript, and for sending us the reviewer comments. We would also like to thank the reviewers, whose thoughtful comments helped us improve a lot our manuscript. We are confident our manuscript is now on a format that is ready for publication. I will paste below the reviewer comments, along with answers to each of them.

Sincerely,

Andre

## REVIEWER COMMENTS:

Reviewer: 1

Reviewer comments: In the manuscript by Oliveira Junior et al., entitled “LC3-associated phagocytosis in macrophage responses to Paracoccidioides spp” the authors investigated the role of LC3- associated phagocytosis (LAP) in the context of Paracoccidioidomycosis spp. infection. First of all, the authors showed by immunofluorescence that LC3 associated phagocytosis is activated in murine macrophages, utilizing both cells lines and primary macrophages. Interestingly, some LC3 detected were in extracellular fungus. To explore the functional relevance of LAP, the authors silenced ATG5 using shRNA and observed an increased fungal burden in infected macrophages, suggesting that ATG5 is important for LAP in Paracoccidioides infection. Furthermore, they inhibited components previously implicated in LAP during other fungal infections and found these were not required in this context. 

The obersvation that LAP is active during Paracoccidioidomycosis is intriguing and of broad interest to the field. Future studies clarifying if LAP is beneficial or detrimental in murine Paracoccidioidomycosis will help us better understand its role in this context. The manuscript is well written but I have some concerns about the methodology and some minor improvements that should be addressed to strengthen the manuscript.

Response: Thank you very much for the positive comments on our manuscript.

Major concerns:

1 – The authors utilized 20 ng/mL of GM-CSF to generate macrophages, although GM-CSF is known to generate these immune cells, other cells such as dendritic cells are found as well (PMID: 26084029 and PMID: 27788572). It is important that the authors clarify whether they performed any purification after cell generation of they utilized the cell pool.

Response: We thank the reviewer for bringing up this important point. GM-CSF stimulation of bone marrow progenitors indeed generates a heterogeneous population. Removing non-adherent cells does decrease the heterogeneity, as pointed in one of the publications the reviewer mentioned (PMID 27788572): around 80% of the adherent cells in this GM-CSF concentration are bone marrow macrophages. 

Action: We have clarified this point explicitly in the Methods section by adding the sentence:

“a condition that results in cultures with approximately 80% macrophages (Siqueira, Fraga et al. 2016). 

[lines 110 - 111]

2 - Another concern regarding the generation of macrophages is that the authors utilized non-adherent plates, but these macrophages are well-known for being adherent, I wonder if the authors had a reason for non-treated plates or if they checked cell viability/adherence/differentiation (PMID: 26084029 and PMID: 27788572).

Response: We used plates that had not been vacuum plasma treated, but this does not mean that these plates are non-adherent. To the contrary, macrophages do adhere firmly to these plates and have to be enzymatically detached, whereas dendritic cells typically stay in suspension or loosely attached. The use of non-treated culture plates is a well-established approach in GM-CSF differentiation protocols. Cell viability was consistently above 95% as assessed by trypan blue exclusion, and the differentiated macrophages exhibited typical morphology with large cytoplasm and prominent pseudopodia.

Action: We added the following phrase to the Methods section: “Cell viability was consistently >95% as measured by trypan blue exclusion.” [lines 118 - 119]

Minor concerns:

1 – Why authors used anti-human LC3 if murine macrophages were being used?

Response: The manufacturer’s datasheet states that this antibody has been validated for detection of LC3 proteins from mouse, rat, and human origin. This is not unexpected, given how conserved LC3 is in both species. The immunofluorescence pattern we observed is consistent with that.

Action: None.

2 – Figure 3 caption could be improved.

a. There is no mention of what A, B, C, D, and E stands for.

Response: We agree the figure caption was not clear.

Response: We have modified the figure caption to explain what the labels mean [lines 354 - 367].

b. EGFP controls should be described to the methods section as well.

Response: We agree a better description is necessary not only in the methods section, but in the rest of the manuscript as well.

Action: We have added to the Methods section the following sentence: “As negative control, we used a similar vector encoding an shRNA that targets the enhanced green fluorescent protein (EGFP) gene, a sequence that is not present in the cell.” [lines 123 - 125].

We have also improved the description of these controls on the Results section Results [lines 235 – 237] and Figure 3 legend [lines 357 – 358].

c. It should be explicit in 3B that J774 is the non-silenced control cells.

Response: We agree.

Action: As described above, we have thoroughly improved the legend to figure 3. The sentence we added to address this reads: “ATG5-silenced clones A and B are compared to negative controls, including untransduced macrophages (J774) and cells in which the shRNA target is not present (EGFP)” [lines 362 - 364].

3 – In lines 237-239 it should say that the reduced antifungal activity was only achieved in clone B, since clone A had no significant difference.

Response: We agree.

Action: We have reformulated that section of the text to read: “A significant reduction in antifungal activity in comparison with the non-silenced control was observed in ATG5-silenced clone B, but not on clone A. Targeting a sequence that does not exist on the macrophage, EGFP, had not effect, confirming that the reduced antifungal activity is specific for ATG5 silencing in clone B (Figure 3B).” [lines 233 - 238].

4 – In line 264 the authors suggest that vomocytosis might occur. If possible, providing representative images or videos would strengthen this claim.

Response: We agree it would be great to have videomicroscopy results to measure non-lytic exocytosis (vomocytosis). We did just that on a previous work with Cryptococcus neoformans (Nicola et al., 2012), but that required a specialized spinning disk confocal microscope and two years of work. However interesting that subject is, it is beyond the scope of this work. We chose to keep this speculation on the discussion section, but added a sentence to clarify that it is just speculation.

Action: We added the following sentence to the Discussion: “testing this tantalizing speculation, however, is beyond the scope of the present work.” [lines 266 - 267].

5 – In line 271 the authors say there is a modest reduction in fungal burden, but that is the opposite as the figure shows.

Response: We thank the reviewer for picking up this mistake.

Action: We have corrected the text to accurately reflect our findings: “However, the modest reduction in antifungal activity observed, even in ATG5-silenced macrophages, suggests that LAP operates as part of a broader immune response rather than acting as the sole antifungal process.” [Line 272].

6 – In line 301 a reference should be added to support the statement regarding β-glucan.

Response: We agree.

Action: We have added two references to that sentence, which now reads: “Differences in fungal cell wall composition, particularly in surface exposure of α- and β-glucans and other pattern recognition receptor ligands (San-Blas, San-Blas et al. 1977, Puccia, Vallejo et al. 2011), may underlie these mechanistic differences.” [Lines 297 - 300]. [Image of fungal cell wall structure] 

## Reviewer: 2

Reviewer comments: The manuscript by Oliveira Junior et al. presents interesting and relevant findings on the role of LC3-associated phagocytosis (LAP) in the macrophage response to Paracoccidioides spp., contributing to our understanding of host-pathogen interactions in paracoccidioidomycosis (PCM). The demonstration of LC3 recruitment to phagosomes and the use of RNA interference to highlight the importance of ATG5 are strengths of the study. Moreover, the observation that canonical LAP signaling pathways (Syk and NADPH oxidase) are not required in this context adds novelty and suggests distinct mechanisms of fungal recognition.

We thank the reviewer for pointing out our work’s strength.

However, the manuscript requires clarification and revision in several areas before it can be considered for publication.

Major Concerns

1. The manuscript relies heavily on immunofluorescence analysis. However, many of the presented images do not clearly distinguish between phagocytosed fungi and those merely adhering to macrophage surfaces. The microscopy system used may not be optimal for resolving this distinction. In Figure 1, it is unclear what the authors are highlighting. The LC3 staining is macrophage-associated, yet based on the presented images, there seems to be LC3 signal external to the phagocytic cells. In some images (e.g., Fig. 1B and 1C), fungal structures are labeled both on the macrophage-interacting and non-interacting sides. A possible solution would be to stain the actin cytoskeleton of macrophages with phalloidin to more clearly delineate cell boundaries and confirm intracellular localization. 

Response: We share the reviewer’s wonders in this case, and have done so since our first LC3 immunofluorescence experiments with other fungi (Cryptococcus neoformans and Candida albicans) almost 20 years ago. In all of them, we have observed similar LC3 recruitment to the surface of fungi that are clearly inside macrophages, but also to fungi that appear to be outside any phagocyte. This happened reproducibly in tests done by several researchers, in two different labs in different countries. This puzzling observation is not a technical issue, as high aperture oil immersion objectives and constrained iterative deconvolution allow optical sectioning at the very limit of resolution in conventional fluorescence microscopy. Thus, repeating the experiment with fluorescent phalloidin would bring no additional insights.

We have speculated in the present manuscript that this could be a result of non-lytic exocytosis, something that we have shown directly in our previous work with other fungi (Nicola et al., 2012). As explained to the first reviewer, however, testing so takes years and is the subject of future work, not this manuscript.

Action: We have modified the following sentence in the discussion to clarify this issue:

“This could represent non-lytic exocytosis (vomocytosis) of the phagosomal contents, a process previously reported in interactions between Paracoccidioides spp. and amoebae (Gonçalves, Ferreira et al. 2019); testing this speculation, however, is beyond the scope of the present work.” [Lines 264-267] 

2. An important methodological concern that should be addressed is whether the anti-LC3 antibody used in the study might cross-react with LC3 homologs in Paracoccidioides spp. Could the observed labeling correspond, at least in part, to fungal LC3-like proteins? The authors should discuss this possibility and ideally demonstrate that the LC3 signal is indeed specific to host LC3 and not a fungal ortholog.

Response: We agree with the reviewer. Using exactly the same staining protocol but with Paracoccidioides spp. cells results in no fluorescence at all (see new Figure S1B). This is expected, as the fungal LC3 homolog (Atg8) is very different from the mammalian counterpart that was used to raise the commercial antibodies we used.

Action: We added to Figure S1 an additional panel, and also added to the Materials and Methods section the following sentence: “Additional experiments with no macrophages, only Paracoccidioides spp. cells, showed that the LC3 antibody does not bind to the fungal cell (Figure S1B).” [Lines 198 – 199].

3. Similar concerns arise with the data shown in Figure 4 using pharmacological inhibitors of Syk and NADPH oxidase. The images do not provide convincing evidence regarding the precise localization of LC3 staining and whether the signal corresponds to phagocytosed fungi or extracellular particles.

Response: We agree with the reviewer regarding the issue of differentiating if the fungi surrounded by LC3 are inside or outside the macrophages, as discussed above. We respectfully disagree with the statement that the images are not convincing evidence of precise localization of LC3 staining. We did see very reproducible localization of LC3 around fungal cells. This is the same pattern that we and others have observed before on multiple fungal species that were co-incubated with macrophages.

Action: None.

4. Interpretation of Figure 3B. The authors conclude that ATG5 knockdown leads to reduced macrophage antifungal activity. However, an alternative explanation, enhanced fungal uptake, cannot be ruled out based on the current data. To support their interpretation, the authors should present phagocytic index data under the same conditions used in this assay to clarify whether the difference in CFU reflects altered killing or increased phagocytosis.

Response: We appreciate this thoughtful comment. It is important to note, though, that as we described in the materials and methods, at no moment between adding fungi and collecting the well contents did we wash the wells. This means that higher phagocytosis does not explain the observed difference in CFU, as all fungi that were added to each well remained there until collected after 24 h.

Action: None.

5. Discussion. The hypothesis that differences in the fungal cell wall account for variations in LAP induction is intriguing but underdeveloped. The authors should support this claim with references to existing literature and, if possible, provide data on basic fungal surface features under the conditions used.

Response: We are glad the reviewer found this intriguing, we also did. However, we have not pursued this avenue further right now. Although it certainly will be the foundation for future experiments, we argue that our present manuscript, a first demonstration that LC3-associated phagocytosis plays a role in the interaction of macrophages and Paracoccidioides spp, is a full story as is. We agree with the reviewer’s suggestion to support this speculation with more references, so we have added two references about the Paracoccidioides cell wall and its effects on virulence. We did not, however, develop this any further because we have not experimentally addressed the issue.

Action: As explained in response to the first reviewer, we have updated a sentence in the discussion to: “Differences in fungal cell wall composition, particularly in surface exposure of α- and β-glucans and other pattern recognition receptor ligands (San-Blas, San-Blas et al. 1977, Puccia, Vallejo et al. 2011), may underlie these mechanistic differences.” [Lines 297 - 300].

This is a promising and potentially impactful study. However, to reach its full potential, the manuscript requires revisions to enhance methodological clarity, support key conclusions, and improve figure interpretation. I encourage the authors to address the issues above in a revised version.

Minor Points

a. The multiplicity of infection (MOI) used in the phagocytosis assays is not specified and should be clearly stated.

Response: We thank the reviewer for noting this omission.

Action: We have updated the Materials and Methods section to clarify we used MOI of 1 in all experiments [lines 161 - 162].

b. Figure legends are insufficiently descriptive and occasionally include interpretations of results rather than objective descriptions. These should be revised for clarity and accuracy.

Response: We agree.

Action: We have carefully revised the figure legends.

---

## [Reviewer Report · REVIEWERS COMMENTS]

## REVIEWER #1

I thank the authors for addressing all of my comments. The manuscript has improved and I have no further concerns.

## REVIEWER #2

The authors have satisfactorily addressed all the concerns raised in my previous review. The revised version of the manuscript adequately incorporates the suggested clarifications and improvements. I have no further comments.